# Integrated Bioinformatics Analysis Reveals Novel miRNA as Biomarkers Associated with Preeclampsia

**DOI:** 10.3390/genes13101781

**Published:** 2022-10-02

**Authors:** Mariarita Brancaccio, Caterina Giachino, Assunta Maria Iazzetta, Antonio Cordone, Elena De Marino, Ornella Affinito, Maria Vivo, Viola Calabrò, Alessandra Pollice, Tiziana Angrisano

**Affiliations:** 1Department of Molecular Medicine and Medical Biotechnology, University of Naples Federico II, 80131 Naples, Italy; 2Department of Biology, University of Naples Federico II, 80126 Naples, Italy; 3Department of Chemistry and Biology, University of Salerno, 84084 Fisciano, Italy

**Keywords:** bioinformatics, preeclampsia, microRNA, biomarkers, epigenetic, therapeutic targets

## Abstract

Preeclampsia is a leading cause of perinatal maternal-foetal mortality and morbidity. This study aims to identify the key microRNAs (miRNA) in preeclampsia and uncover their potential functions. We downloaded the miRNA expression profile of GSE119799 for plasma and GSE177049 for the placenta. Each dataset consisted of five patients (PE) and five controls (N). From a technical point of view, we analysed the counts per million (CPM) for both datasets, highlighting 358 miRNAs in common, 78 unique for plasma and 298 unique for placenta. At the same time, we performed an expression differential analysis (|logFC| ≥ 1|and FDR ≤ 0.05) to evaluate the biological impact of the miRNAs. This approach allowed us to highlight 321 miRNAs in common between plasma and placenta, within which four were upregulated in plasma. Furthermore, the same analysis revealed five miRNAs expressed exclusively in plasma; these were also upregulated. In conclusion, the in-depth bioinformatics analysis conducted during our study will allow us, on the one hand, to verify the targets of each of the nine identified miRNAs; on the other hand, to use them both as new non-invasive biomarkers and as therapeutic targets for the development of *personalised* treatments.

## 1. Introduction

It has been reported that, in countries with advanced economies, 10–20% of pregnancy is affected by some form of hypertension [1]. The exact incidence of preeclampsia (PE) is not known, but it is believed that it can be around 5–8% [2]. Preeclampsia also represents the basis of 15–20% of maternal mortality cases [3] and constitutes one of the leading causes of perinatal mortality and morbidity [4]. It is usually diagnosed based on the de novo onset of hypertension and proteinuria [5]. The lack of reliable methods for early detection limits the opportunities for prevention, diagnosis and timely treatment.

MicroRNAs (miRNAs) are small endogenous single-stranded non-coding RNA molecules found in the transcriptome of plants, animals and some DNA viruses [6,7,8]. These are polymers encoded by eukaryotic nuclear DNA about 20–22 nucleotides long and are mainly active in the regulation of gene expression at the transcriptional and post-transcriptional levels [6,7,8]. The miRNAs are incorporated into the RNA-induced silencing complex (RISC) and induce gene silencing by overlapping with complementary sequences present on target messenger RNA (mRNA) molecules [6,7,8]. In addition, it is known that miRNAs are responsible for the epigenetic regulation of several molecules [9]. This link leads to translation repression or degradation of the target molecule [6,7,8].

A lot of evidence regarding the role of miRNAs in preeclampsia has been accumulated in the last 10 years [10]. Differentially expressed miRNAs are characteristic of both mild and severe PE [10]. In many cases, they target signalling pathway-related genes that result in altered processes directly involved in PE [10].

High-throughput platforms such as Next Generation Sequencing technology (NGS) are increasingly popular for miRNA and gene expression analysis in PE. For example, Mavreli and collaborators use an NGS approach, highlighting that miR-23b-5p and miR-99b-5p are downregulated in plasma of PE compared to controls [11]. In the same work, the authors highlighted that the targets of these microRNAs were associated with glycometabolism and the immune response involved in the pathogenesis of PE [11].

On the other hand, Vashukova and co-authors, using Ion Tor-rent next-generation sequencing technology, identified a total of 22 miRNAs that included 16 miRNAs previously known to be associated with PE and six novel miRNAs [12]. Among the six new miRNAs, four were upregulated (miR-518a, miR-527, miR-518e and miR-4532) and two were downregulated (miR-98 and miR-135b) in PE placentas compared to controls [12].

The results currently reported in the literature suggest that PE is associated with specific alterations in the expression pattern of placental miRNAs.

We decided to compare miRNAs expressed between plasma and placenta for the following reasons. First of all, the placenta is a deciduous, and therefore temporary, organ that forms in the uterus during pregnancy. The placenta is responsible for nourishing, protecting and supporting fetal growth. The placenta is common to the pregnant woman and the fetus; a part of it has maternal origins (constituted by the modified or deciduous uterine endometrium), while the remainder has fetal origins (formed by the chorionic villi). Therefore, the placenta represents the fetus’s roots in the mother’s soil. Secondarily, plasma is the liquid component of the blood, in which the corpuscular elements (red blood cells, white blood cells and platelets) are suspended. Finally, the mother’s immune system during pregnancy must be tolerant of the fetus during gestosis; to do this at the plasma level, chemokines, miRNAs, and cytokines that do not allow premature abortions are released.

Consequently, the identification of common biomarkers can be of help in PE monitoring.

In light of this, our work aimed to compare, through a bioinformatics approach, the gene expression profile of miRNAs coming from two databases. The databases used were GSE119799 for plasma [11] and GSE177049 for the placenta to discover the miRNAs and key genes that contribute to the pathology of PE and thus provide new insights into potential biomarkers for prognosis and therapeutic strategies of PE.

## 2. Materials and Methods

### 2.1. Data Collection and Processing

The following miRNA datasets were downloaded from the Gene Expression Omnibus: GSE177049 containing miRNA data from ten placental samples (*n* = 5 control (N); *n* = 5 patient with preeclampsia (PE)) and GSE119799 [11] containing miRNA data from ten plasma samples (*n* = 5 control (N); *n* = 5 patients with preeclampsia (PE)).

For placental samples, only processed expression data (CPM) for 794 miRNAs were available and obtained as described in https://www.ncbi.nlm.nih.gov/geo/query/acc.cgi?acc=GSE177049 (accessed on 27 July 2022). Briefly, miRDeep2 is a software package for identification of novel and known miRNAs in deep sequencing data; therefore, it was used to quantify the expression level of known mature miRNAs curated in the human miRBase database. The miRNA-seq was analysed by R/Bioconductor package Edge R [13], which employs a negative binomial generalised linear model with likelihood ratio test (glmLRT) to compare the miRNAs expression level between two conditions. Only those miRNAs (*n* = 656) expressed at counts per million (CPM) above 0.5 in at least 90% of the samples.

For plasma samples, raw counts were available for 4483 miRNAs, out of them 1753 were novel miRNAs and 2730 known miRNAs. Out of them, to best compare plasma and placental samples, we filtered out the novel miRNAs, and we retained only those known miRNAs (*n* = 436) expressed at counts per million (CPM) above 0.5 in at least 90% of the samples After a TMM normalisation, differential expression was assessed with glmLRT method implemented in R/Bioconductor package Edge R [13].

For both datasets, only miRNAs genes with |logFC| ≥ 1 and an adjusted *p*-value (FDR) ≤ 0.05 were defined as differentially expressed.

Venn diagram to show overlapping and specific miRNAs was performed by R package VennDiagram.

Heatmap was performed by R package gplots. The count-per-million (CPM) at log2 scale, denoted as logCPM, was computed for visualisation of miRNA abundance in heatmap-clustering analysis.

### 2.2. Prediction of miRNAs’ Targets

To identify the targets of the collected miRNAs, we used three different tools: (1) miRDB, an online database for the prediction of the miRNA targets [14,15]; (2) RNA22 version 2.0, software for miRNA target predictions [16], and TargetScan 8.0, a web server of miRNA target predictions [17,18,19,20,21,22].

We first searched for targets in miRDB using 60–100 as a range for the Target Score [14,15]. The targets identified with miRDB were then analysed in RNA22 version 2.0 using the following filters: base pair min value = 12, folding energy max value (Kcal/mol) = −12 and *p*-value < 0.05. Lastly, only the targets identified in RNA22 version 2.0 were validated in TargetScan 8.0 taking the total context ++ score as the prediction truthfulness values [18].

### 2.3. Functional Enrichment Analysis

Functional enrichment analysis of target genes was performed by using Kyoto Encyclopedia of Genes and Genomes (KEGG) database implemented in miRNet 2.0 [23]. Statistical significance was evaluated by the hypergeometric test. Only pathways with FDR ≤ 0.05 were considered as significative.

## 3. Results

### 3.1. The Plasma and Placental miRNoma Landscape in Preeclampsia

miRNoma landscape in preeclampasia was analysed in two public datasets: GSE119799 for plasma [11] and GSE177049 for the placenta. After appropriate filtering and normalisation procedures, we obtained a total 656 miRNAs in placenta and 436 in plasma (see Section 2 for details). As shown in Venn diagram, 358 miRNAs were in common between placenta and plasma while the remaining were specific of each tissue (see Figure 1).

### 3.2. Comparison of miRNAs Expression Profiles in a Woman with PE

To clarify the role of miRNAs in PE, we have focused on 358 miRNAs in common between placenta and plasma.

Focusing on plasma, among these 358 miRNAs, we extracted four differentially expressed miRNAs (|logFC| ≥ 1 and FDR ≤ 0.05). All of them were upregulated in comparison with the control (Figure 2). The same miRNAs resulted in being non-differentially expressed between control and preeclampsia patients in placenta. A comparison of the expression levels of these four common miRNAs in plasma and placenta is reported in Figure 3.

Furthermore, we also evaluated the expression levels of plasma-specific miRNAs. We found five plasma-specific differentially expressed miRNAs, all of them upregulated with respect to normal samples (Figure 4).

### 3.3. Annotation of miRNAs Involved in the Pathogenesis of Preeclampsia and Their Targets

To investigate the role of the 4 miRNAs found in common between plasma and placenta, we carried out a systematic analysis identifying the targets and the role of these miRNAs in the pathogenesis of preeclampsia (see Table 1).

The same type of analysis was performed for the 5 miRNAs unique to plasma (see Table 2).

Following this analysis, we have shown that 1 (hsa-mir-378e) of the 4 miRNAs identified had never been reported as biomarkers for preeclampsia or diseases related to it, but as a marker of pathologies linked to the endometrium.

On the other hand, for the remaining 3 (hsa-mir-1246, hsa-mir-203a-3p and hsa-mir-378c), the targets that were activated and/or repressed during the pathophysiology of preeclampsia were also noted (see Table 1). As reported in Table 1, most of the targets appear to be involved in the process of placentation and inflammation.

Among these 5 miRNA, 2 (hsa-miR-708-3p and hsa-miR-4508) are not known as possible miRNA involved in the pathophysiology of preeclampsia, but disorders such as cancer and autoimmune diseases are found (Table 2). Instead, the other 3 miRNAs (hsa-miR-4500, hsa-miR-4516 and hsa-miR-4497) have been found in processes that may occur during preeclampsia such as intrauterine growth retardation or preterm birth (Table 2).

### 3.4. Analysis of the Predicted miRNA Targets

To shed light on the role of the 3 unidentified miRNAs (hsa-mir-378e, hsa-miR-708-3p and hsa-miR-4508) using bioinformatics analysis that did not have known targets to the pathophysiological process of preeclampsia, we predicted through miRDB using 60–100 as a range for the Target Score [14,15,16,17,18,19,20,21,22].

Following this analysis, we identified 310 targets for hsa-mir-378e, 457 targets for hsa-miR-708-3p and 125 targets for hsa-miR-4508 (Figure 5 and Appendix A). After that, we merged the obtained results with RNA22 version 2.0 using the following filters: base pair min value = 12, folding energy max value (Kcal/mol) = −12 and *p*-value < 0.05. In this case, 57 common targets for hsa-mir-378e, 2 common targets for hsa-miR-708-3p and 15 common targets for hsa-miR-4508 were found (Figure 5 and Appendix A).

Finally, to validate the identified targets, we used TargetScan 8.0, taking the total context ++ score as the prediction truthfulness values [18] (Appendix A). In particular, we highlighted 41 targets for hsa-mir-378e, 2 targets for hsa-miR-708-3p and 13 targets for hsa-miR-4508 (see Figure 5 and Appendix A).

### 3.5. Functional Enrichment Analysis

To better investigate functional pathways where target genes are involved, we performed an enrichment analysis by using a KEGG database implemented in miRNet 2.0 [23] (Figure 6). Interestingly, we found three pathways potentially associated with preeclampsia: MAPK signalling pathway (FDR = 0.006), TGF-**β** signalling pathway (FDR = 0.011), Insulin signalling pathway (FDR = 0.020) and VEGF signalling pathway (FDR = 0.031) (Figure 6).

## 4. Discussion

PE is a multisystem disorder specific to pregnancy, and a deficiency in our knowledge of the exact aetiology and pathogenesis of this human disorder restricts the ability to treat this disease [46]. Therefore, the identification of molecular targets to build a panel of biomarkers involved in the pathophysiology of PE is crucial to develop more effective diagnostic and therapeutic strategies.

At the same time, it is now known that miRNAs play a fundamental role as next-generation biomarkers [47,48,49,50,51,52,53,54,55,56]. The miRNAs being measurable through biological fluids and regulating numerous cellular targets represent “the gold standard of biomarkers” in pathologies in which a rapid prognosis and constant monitoring are required [47,48,49,50,51,52,53,54,55,56].

In light of this, our study focused on identifying miRNAs that can be used as biomarkers in preeclampsia.

First of all, through deep bioinformatics analysis, we have identified the miRNAs in common with the placenta and plasma. The search for stable miRNAs in the two tissues mainly involved in the pathophysiology of preeclampsia aims to employ these miRNAs as specific and sensitive biomarkers. Indeed, the biomarkers used in diagnosing and monitoring multifactorial disorders must be peculiar [47,48,49,50,51,52,53,54,55,56].

We then focused our attention on up and down miRNAs regulated between the placenta and plasma. In particular, the placenta and the foetus are connected by the umbilical cord; the placenta represents the sustenance system of the foetus because it functions as an exchange point between maternal and foetal blood.

Recently, Chang and co-workers showed that placental miRNA trafficking mainly to the maternal circulation and that maternal miRNA could transit to the placenta and even into the foetal compartment. These findings define an extraordinary miRNA-based non-hormonal means of communication between the placenta and the foetus-maternal compartments [57]. Consequently, the trafficking between maternal and foetal blood is of fundamental importance for monitoring both the state of health of the mother and the foetus [52]; therefore, the identification of stable miRNAs in common between the placenta and the blood helps monitor PE and the correct development of the foetus.

Furthermore, it is now known that placental cell disorders are closely associated with adverse pregnancy outcomes and the manifestation of PE. At the same time, numerous miRNAs have been found in the human placenta and growing evidence is revealing their role as protagonists in regulating placental cell behaviours. Indeed, recent studies indicate that placenta-derived miRNAs can be released into the maternal circulation by encapsulating them in exosomes and potentially targeting various maternal cells [58].

In our case of the 4 miRNAs in common between placenta and plasma that we highlighted, 3 (hsa-mir-1246, hsa-mir-203a-3p and hsa-mir-378c) were known actors in the pathophysiology of PE [24,25,26,27,28,29,30]; instead, the other 1 (hsa-mir-378e) has never been indicated as a possible biomarker of PE.

In addition, the targets prediction analysis of hsa-mir-378e allowed us to highlight two interesting targets that appear to be involved in the pathophysiology of PE: insulin-like growth factor-1 (IGF-1) and mitogen-activated protein kinase 1 (MAPK-1) [59,60,61,62,63].

Plasma IGF-1 levels are known to correlate with the severity of preeclampsia [59,60,61,62]. In fact, in the study conducted by Ingec and collaborators [60], IGF-1 and insulin-like growth factor binding protein-1 (IGFBP-1) were measured in groups with different degrees of PE (20 mild pre-eclamptic, 20 severe pre-eclamptic, 20 eclamptic patients and 20 healthy pregnant women in the third trimester) [60].

The authors pointed out on IGF-1 levels were lower in pre-eclamptic and eclamptic patients than in controls, while IGFBP-1 had an opposite trend [60].

The deficiency of IGF-1 could cause harm to the foetus; in fact, Halhali and co-authors [62] have shown that, in women suffering from preeclampsia, there is a decrease in IGF-1 levels also in the cord blood, influencing the growth of the foetus [62].

On the other hand, it is known that impaired trophoblast invasion partly modulated by abnormal MAPK1/ERK2 signalling played important roles in the pathological process of preeclampsia [63].

At the same time, within the 5 miRNAs unique to plasmas, we have brought to light 2 miRNAs (hsa-mir-708-3p and hsa-mir-4508) which do not appear to be noted as actors of PE [32,33,34,35].

For hsa-mir-708-3p, we have identified 16 targets, but of particular importance is the fibroblast growth factor receptor 2 (FGFR2). In fact, recent studies have shown that specific mutations of FGFR2 can be the substrate for the emergence of PE [64]. Furthermore, it has been shown that the methylation status of FGFR2 influences the phenomenon of placentation and especially sex-specific preterm births [65].

In addition, for hsa-mir-4508, we have identified 18 targets; among these, of particular importance is solute carrier family 43, member 3 (SLC43A3) and solute carrier family 12 (potassium/chloride transporter), and member 4 (SLC12A4), both involved in the regulation of prolactin levels in the amniotic fluid and therefore responsible for foetal development [66].

Finally, during functional enrichment analysis, on the one hand, we confirmed two signalling pathways that emerged during the search for specific targets: MAPK and Insulin signalling pathway [59,60,61,62,63]; on the other hand, we have shed light on two other extremely interesting signalling pathways for preeclampsia: TGF-β and VEGF [67,68].

Normal human pregnancy depends on the physiological transformation of spiral arteries by invasive trophoblasts. Preeclampsia is associated with impaired trophoblast invasion and spiral artery transformation. Recent studies have suggested that TGF-β is overexpressed in the placenta of PE patients and that this may be responsible for failed trophoblast invasion [67].

Moreover, there is additional experimental evidence that supports the hypothesis that interference with VEGF/PlGF signalling could mediate endothelial dysfunction in preeclampsia. VEGF is important in the stabilisation of endothelial cells in mature blood vessels. VEGF is particularly important in the health of the fenestrated and sinusoidal endothelium found in the renal glomerulus, brain, and liver organs severely affected in preeclampsia. VEGF is highly expressed by glomerular podocytes, and VEGF receptors are present in glomerular endothelial cells [68].

The meticulous bioinformatic analysis and target prediction of miRNAs not yet known as protagonists of the pathophysiological process of PE have allowed us to shed light on new biomarkers that could also represent new therapeutic targets for new therapies. Having highlighted nine upregulated miRNAs in plasma, four of which are also expressed in the placenta, can help in the development of rapid and non-invasive tests to monitor preeclampsia during gestation without having to wait for birth to be able to access tests on the placenta and/or on the foetus.

## 5. Conclusions

The data shown here confirm that miRNAs play a fundamental role in the pathophysiology of preeclampsia [69]. In addition, the in-depth bioinformatics analysis carried out on the validation of new targets allowed us to unearth new biomarkers. In conclusion, our study allowed us to lay the foundations for the development of a panel of biomarkers that can be used both as prognostic factors and as monitoring factors for both pathology and any drug therapies.

## Figures and Tables

**Figure 1 genes-13-01781-f001:**
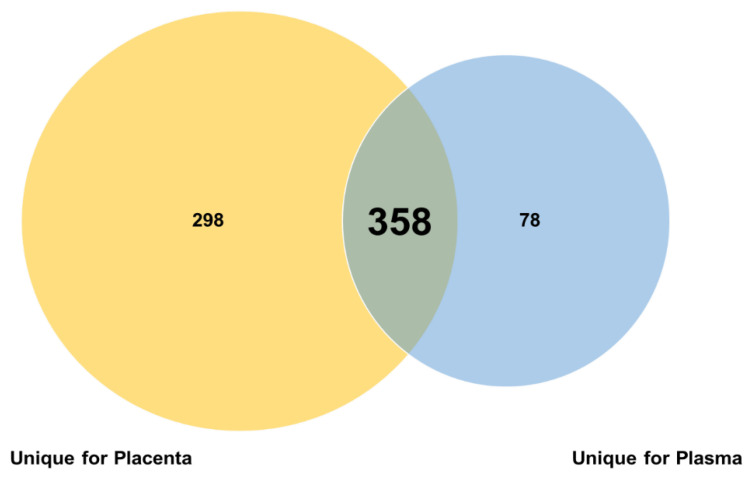
Venn diagram of overlapping and specific miRNAs in plasma and placenta.

**Figure 2 genes-13-01781-f002:**
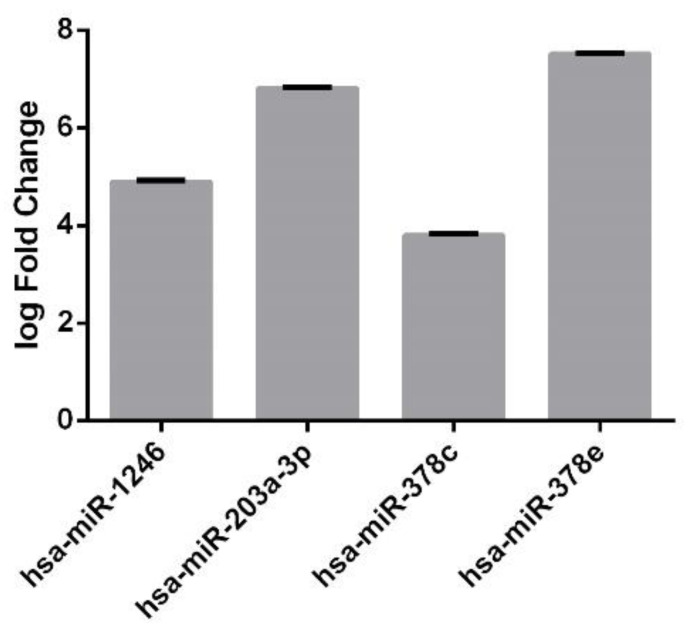
**Differentially Expressed miRNAs in common between plasma and placenta.** Barplot shows the upregulated miRNAs expression levels in plasma. The expression levels are expressed as log Fold Change (log2FC).

**Figure 3 genes-13-01781-f003:**
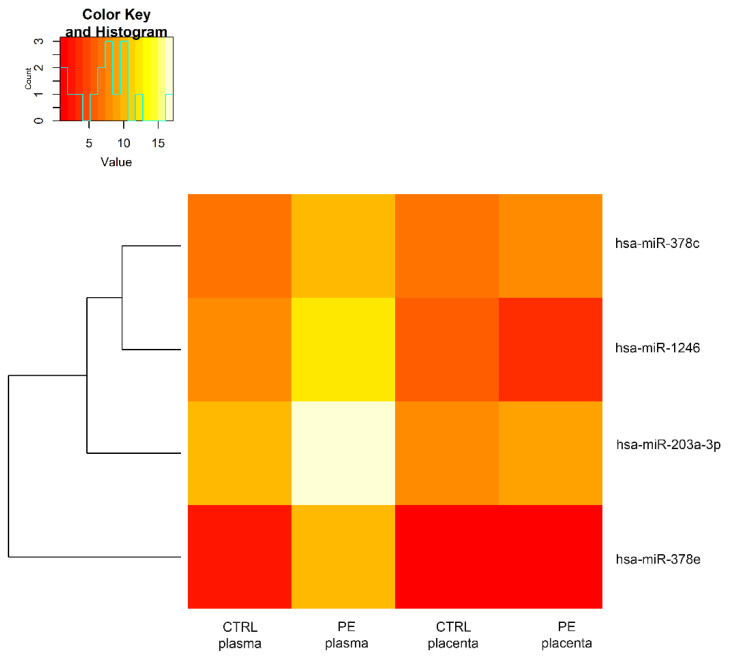
**Heatmap: common miRNAs expression profiles in plasma and placenta**. Heatmap shows the expression levels of each common miRNA in plasma and placenta tissues. The expression level of each miRNA (rows) was averaged within the pre-classified group (column). Values are expressed as log2CPM.

**Figure 4 genes-13-01781-f004:**
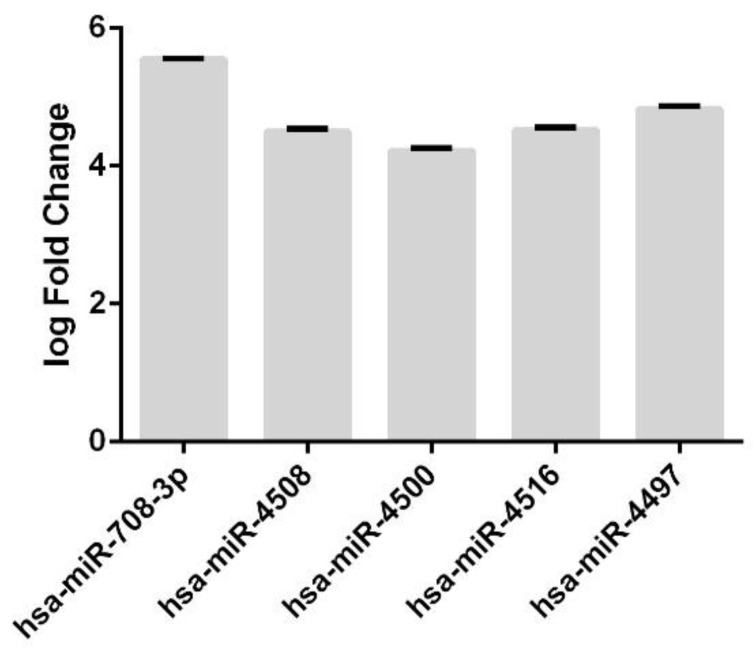
**Differentially Expressed of plasma-specific miRNAs**. Barplot shows the upregulated unique miRNAs in plasma. The expression levels are expressed as log Fold Change (log2FC).

**Figure 5 genes-13-01781-f005:**
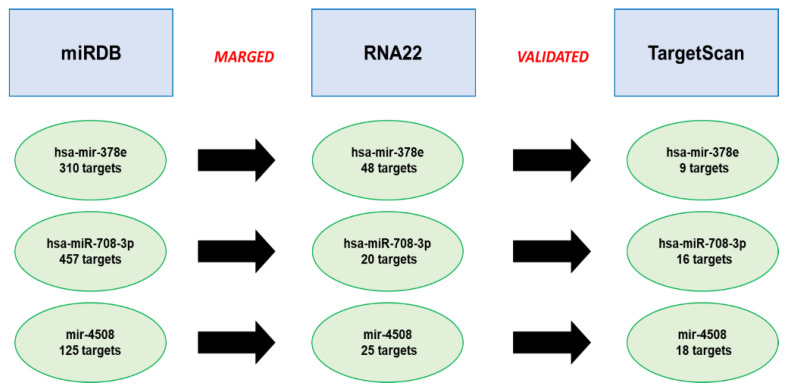
**Predicted targets for preeclampsia miRNAs.** The overlapping target genes were predicted thanks to the online analysis tools: miRDB, RNA22 and TargetScan; in miRDB using 60–100 as a range for the Target Score; in the RNA22 version 2.0 using the following filters: base pair min value = 12, folding energy max value (Kcal/mol) = −12 and *p*-value < 0.05. In TargetScan 8.0, take the total context ++ score as the prediction truthfulness value.

**Figure 6 genes-13-01781-f006:**
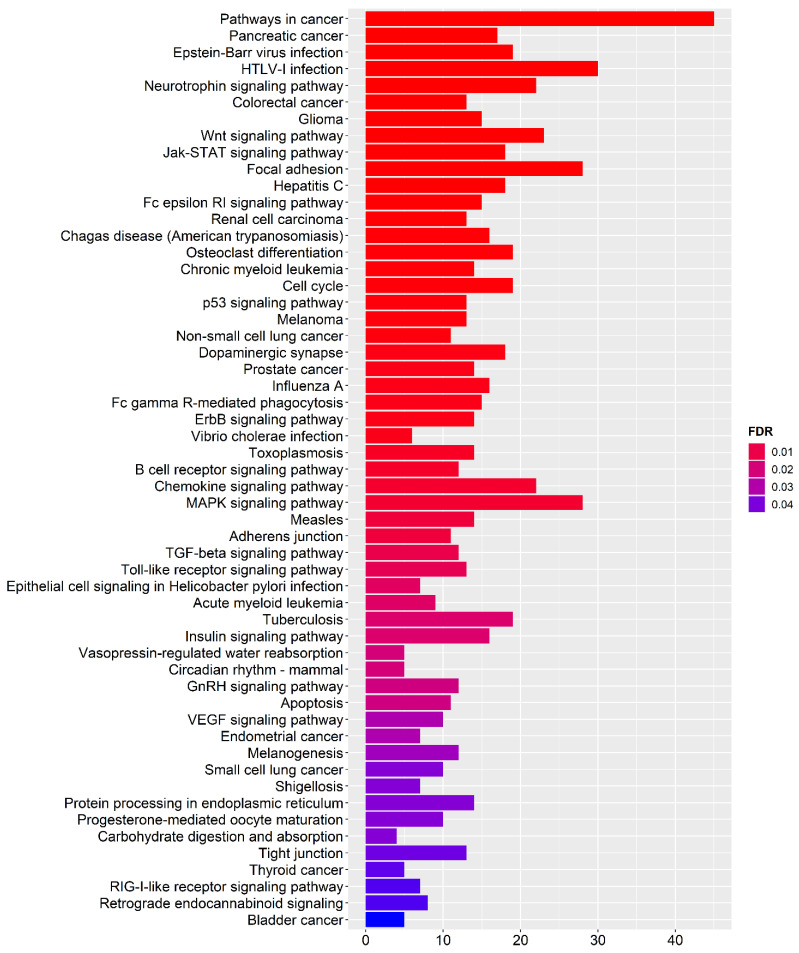
**Pathway Enrichment Analysis**. The *y*-axis represents the enriched KEGG items; the *x*-axis shows the gene count. The color of each bar represents enrichment significance. Only pathways with FDR ≤ are considered as statistically significant.

**Table 1 genes-13-01781-t001:** Role and target of miRNAs in common between plasma and placenta in preeclampsia.

miRNA	Target	References
hsa-mir-1246	interacts with NRF2 promotes the differentiation of the syncytiotrophoblast modulates Jarid2 in human trophoblast differentiation modulates p53, PI3K/AKT, mTOR in glioblastoma	[24,25]
hsa-mir-203a-3p	plays an anti-inflammatory role in PE pregnant women by downregulating the IL24 level isolated in plasma exosomes of women with PE	[26,27,28]
hsa-mir-378c	upregulated in the first trimester of pregnancy in women with PE may lead to the downregulation of genes involved in the HIF-1/VEGF signaling pathway, adversely affecting angiogenesis, implantation and embryo development, and participates in abortion process.	[29,30]
hsa-mir-378e	expressed in the endometrium	[31]

**Table 2 genes-13-01781-t002:** Role and target of upregulated miRNAs in preeclampsia unique for plasma.

miRNA	Target	References
hsa-miR-708-3p	Downregulated in breast cancer, Inhibits metastases through Neuronatin may represent a potential therapeutic target via the ADAM17-GATA/STAT3 axis in idiopathic pulmonary fibrosis	[32,33]
hsa-miR-4508	upregulated in rheumatoid arthritis; is a therapeutic target of the same pathology upregulated in hepatic cancer cells (HepG2)	[34,35]
hsa-miR-4500	isolated in plasma exosomes of women with PE is epigenetically downregulated in colorectal cancer and functions as a novel tumor suppressor by regulating HMGA2 Inhibits Migration, Invasion, and Angiogenesis of Breast Cancer Cells via RRM2-Dependent MAPK Signaling Pathway	[28,36,37]
hsa-miR-4516	potential biomarker for early diagnosis of dust-induced pulmonary fibrosis plays a role in PUVA-induced apoptosis through the downregulation of STAT3/CDK6/UBE2N in keratinocytes downregulated in psoriasis inhibits keratinocyte motility by targeting fibronectin/integrin α9 signalling Biomarker of Premature Ovarian Insufficiency altered in preterm births	[38,39,40,41,42]
hsa-miR-4497	tumor suppressor in laryngeal squamous cell carcinoma via negatively modulation the GBX2 increases the expression and secretion of TNFα induced by LPS in mothers with type 1 diabetes contain upregulated in placentas with selective intrauterine growth restriction	[43,44,45]

## Data Availability

Not applicable.

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
