# Peer review of "Integrated Bioinformatics Analysis Reveals Novel miRNA as Biomarkers Associated with Preeclampsia"

_genes, 2022, doi:10.3390/genes13101781_

Round 1

Reviewer 1 Report

This article uses bioinformatics analyses to identify microRNA(miRNA) biomarkers for preeclampsia (PE). Two miRNA gene expression datasets were used in this study. One is data from plasma samples and the other one is data from placental samples. Twelve overlapping miRNAs were identified for both datasets.

Specific comments

1.      In the abstract, you did not indicate “miRNA” is the abbreviation of “microRNA”, and used it in line 17.

2.      Check this sentence in the abstract “In addition, the significant miRNAs in the placenta, we looked for them 21 in the plasma and found that 12 miRNAs were in common, of which 6 were upregulated and 6 were 22 downregulated.”.

3.      It is not clear why you considered the overlapping miRNAs in plasma and placenta. The differently expressed miRNAs identified in plasma might be suitable to be used as biomarkers for PE because it is easy to have a plasma test. But is it feasible to have a placenta test?

4.      If only plasma analysis is considered, are there any miRNAs that are more significantly different in expression between PE and control groups than the 12 identified miRNAs? What is the p-value ranking of the 12 identified miRNAs among those differentially expressed in the plasma or placental panel?

5.      In the conclusion section, the three sentences can be placed in one paragraph. 

Author Response

Manuscript Number: genes_1862404  

“Integrated Bioinformatics Analysis Reveals Novel miRNA As Biomarkers Associated with Preeclampsia”

Dear Editor,

Thank you for your decision and for the comments about our manuscript. We have really appreciated the reviewers’comments and we believe that they have contributed a lot to improve the quality of our manuscript.

We have addressed all the reviewers’ comments and the responses are appended below.

We hope that in the revised version our manuscript is suitable for publication on Genes

Best regards

Mariarita Brancaccio on behalf of all co-authors

Response to reviewers’ comments

Reviewer 1#

This article uses bioinformatics analyses to identify microRNA(miRNA) biomarkers for preeclampsia (PE). Two miRNA gene expression datasets were used in this study. One is data from plasma samples and the other one is data from placental samples. Twelve overlapping miRNAs were identified for both datasets. 

Specific comments

  • In the abstract, you did not indicate “miRNA” is the abbreviation of “microRNA”, and used it in line 17.

Response

Thank you for pointing out this oversight; we have changed the text and reported the abbreviation in breckets. The change is highlighted in yellow.

  • Check this sentence in the abstract “In addition, the significant miRNAs in the placenta, we looked for them 21 in the plasma and found that 12 miRNAs were in common, of which 6 were upregulated and 6 were 22 downregulated.”

Response

Thanks for the suggestion; we are in agreement with you. In light of this, we have rewritten the abstract; we hope to have improved it.

  • t is not clear why you considered the overlapping miRNAs in plasma and placenta. The differently expressed miRNAs identified in plasma might be suitable to be used as biomarkers for PE because it is easy to have a plasma test. But is it feasible to have a placenta test?

Response

Thanks for the question. To date, there are no rapid tests that exploit circulating miRNAs and obviously, there are no rapid in vivo tests on the placenta. In fact, to date, the moment of give birth is expected to cover what happened afterwards.

In light of this, our work has shown that plasma-up-regulated miRNA could be used to develop a non-invasive method that helps to monitor the onset and progression of preeclampsia during the nine months; at the same time these miRNAs could be used as therapeutic targets.

To clarify this we have modified the manuscript lines 272-275.

  • If only plasma analysis is considered, are there any miRNAs that are more significantly different in expression between PE and control groups than the 12 identified miRNAs? What is the p-value ranking of the 12 identified miRNAs among those differentially expressed in the plasma or placental panel?

Response

Thanks for watching. In accordance with what you pointed out, we carried out an in-depth analysis of the datasets that we reported both in materials and methods and in results. We hope we have improved the quality of the manuscript.

  • In the conclusion section, the three sentences can be placed in one paragraph.

Response

Thanks for the suggestion; we are in agreement with you. In light of this, we have modified the text.

Reviewer 2#

In this study, the authors sought to identify novel microRNA biomarkers associated with preeclampsia through bioinformatics analysis of two publically available datasets made from small RNA sequencing of microRNAs from the placenta and plasma (GSE177049 and GSE119799 respectively). The authors identified microRNAs that were differentially expressed during preeclampsia in both the placenta and plasma, and then identified mRNA targets of 5 previously unidentified microRNAs. This useful study could help identify biomarkers of preeclampsia.

Overall, the use of 2 different datasets to identify common microRNA biomarkers of preeclampsia in different tissues is a strength of this study. The authors also did an excellent job of reviewing the literature with respect to the specific microRNAs in question. However, from reading the manuscript it is not clear whether their bioinformatics approach is sufficiently robust to identify altered microRNAs during preeclampsia. In addition, some parts of the manuscript were challenging to understand. I would recommend additional editing of the text to improve clarity for the reader.

Specific comments

(1) How were the datasets pre-processed?

The authors accessed two publically available datasets (GSE177049 and GSE119799) from Gene Expression Omnibus (GEO). GEO provides both raw sequencing files in the ‘fastq’ format, as well as read count gene expression matrices. From reading the manuscript, it is unclear whether the authors downloaded raw fastq files and performed their own pre-processing, or directly downloaded and used the read count files from GEO. There are different approaches to pre-process this type of microRNA sequencing data that can influence the resulting read counts that are used for differential expression analysis. Pre-processing of this type of data typically includes trimming of low-quality sequences, adaptor removal, alignment to a reference genome, and counting of reads that map to known microRNAs. It would be better for the authors to perform their own pre-processing on both datasets because this would ensure consistent read counts between the two datasets that were used. There are various freely available online tools to perform pre-processing, like Galaxy (https://usegalaxy.org/).

If the authors performed their own pre-processing, this should be clearly described in the methods section.

If the authors directly utilized read count files from GEO, this should be clearly explained and justified in the manuscript – including a brief description of how the two different datasets were originally pre-processed and explanation of why it is valid to make a direct comparison between them.

Response

We agree with the Reviewer’s considerations. We reported in the Materials and Methods: paragraph 2.1 Data collection and processing how we cleaned and analyzed the data sets.

(2) How were the read counts normalized?

There was no description of how the read counts were normalized prior to calculating fold changes and p-values. This is important, because read count normalization drastically affects the statistical significance calculated for each microRNA. It is not valid to simply use counts per million (cpm)/transcripts per million (tpm) etc. for statistical comparisons – this type of normalization is only used for comparing expression levels between different tissues etc., and cannot be used to identify differentially expressed microRNAs. Common tools for normalizing read counts for this type of sequencing data include DESeq2 (http://bioconductor.org/packages/devel/bioc/vignettes/DESeq2/inst/doc/DESeq2.html) and EdgeR (http://bioconductor.org/packages/devel/bioc/vignettes/edgeR/inst/doc/edgeRUsersGuide.pdf), which can take into account library sizes, sequencing depth, batch effects and read distribution.

Response

We agree with the Reviewer’s considerations. We reported in the Materials and Methods: paragraph 2.1 Data collection and processing how we cleaned and analyzed the data sets.

(3) Why were p-values not corrected for the false-discovery rate (FDR)?

A student’s t-test was employed to calculate p-values, with p < 0.05 considered statistically significant (lines 98-100). However, for large -omic datasets like this, it is necessary to apply a FDR correction such as the Benjamini-Hochberg correction. This will produce a more robust assessment of statistical significance. Tools like DESeq2 and EdgeR include apply a FDR correction by default when performing differential expression analysis.

Thanks for the comments. In accordance with the Reviewer’s considerations we calculated the FDR value as a parameter for the statistics. We reported in the Materials and Methods: paragraph 2.1 Data collection and processing.

(4) Principle component analysis.

Principle component analysis (PCA) is a common type of analysis used to assess sources of variation within gene expression data and could be used to visualize the effect size of preeclampsia on microRNA abundance and to identify unwanted sources of variation or batch effects present within the dataset. A PCA plot, or something equivalent, should be included because this will help the reader interpret the robustness of the results.

Response

We agree with the Reviewer’s considerations. Regarding the plasma dataset, raw counts are publicly available; regarding the placenta data, only cpm data are available. We wrote the author to ask for them, but we did not receive any reply. Following that, we processed plasma data by reproducing the workflow of the placenta, as described in Methods, and the differential analysis was done by considering each tissue individually, comparing each group of patients with its own group of normals. In this way, we tried to reduce the technical variability to best compare the two dataset and to extract biological findings as robust as possible.

 (5) Heatmap to visualize microRNA expression.

I would recommend the authors to include a heatmap to visualize the relative abundance of differentially expressed microRNAs across all samples within the datasets. This will help the reader to visualize expression patterns of the microRNAs and interpret the robustness of the association between altered micoRNA abundance with preeclampsia.

Response

Thanks for the comments. In accordance with the Reviewer’s considerations we performed the heatmap (see: Methods lines 110-112, Results lines 145-147 and figure 3).

 (6) Functional enrichment of target genes.

The authors did an excellent job of identifying mRNA targets of microRNAs by using three different databases (miRDB, RNA22, and TargetScan). While the authors did identify target genes implicated in preeclampsia through literature review, it is not clear what types of transcripts were targeted by the microRNAs and how they might be implicated in pathophysiology. I recommend the authors to perform an additional functional enrichment analysis of target genes to identify over-represented pathways and gene ontologies. There are freely available online tools to perform this type of analysis, like DAVID (https://david.ncifcrf.gov/) and EnrichR (https://maayanlab.cloud/Enrichr/).

Response

We agree with the Reviewer’s considerations. For this reason we performed functional enrichment of target genes (see: Methods lines 126-129, Results lines 241-246 and figure 6).

(7) At least one of the datasets used has been previously published (GSE119799 – PMID 32871756). How does this study provide new information, compared to the work that was previously published?

Response

The aim of our work was to highlight common miRNAs between placenta and plasma. In particular, to bring to light miRNAs that could be used as biomarkers of the pathology and at the same time as therapeutic targets. The novelty is that our study highlighted the presence of 10 up regulated miRNAs in plasma, within which 4 are also expressed in the placenta. This scientific evidence is comforting for several reasons:

  1. a) Non-invasive method to assess the state of the pathology step by step over 9 months (simple arterial blood sampling) in vivo placenta biopsy is not a method in use due to the danger of sampling
  2. b) Monitoring of the pathology through specific molecular methods (panel of biomarkers to be linked to the biochemical evaluation of proteinuria)
  3. c) Rapid assessment, there would no longer be the need to wait for delivery.

To clarify our objectives, we have modified both the introduction and the discussion. Changes are shown in yellow.

(8) The specific rationale for comparing placenta and plasma microRNAs should be mentioned in the introduction.

Response

Thanks for the advice. We fully agree with what is requested; therefore, to clarify the rationale in the comparison between plasma and palcenta we have modified the introduction from lines 66 to 78.

(9) Line 122-123: “In addition, we recalculated the p-value of all of them and selected only the significant ones (p-value < 0.05)” – Why were p-values recalculated? Please clarify.

Response

We apologize for being unclear in the description of materials and methods, in light of this we have modified the materials and methods section for both the collection of results and the statistics.

We hope we have improved our manuscript.

All changes in the text are underlined in yellow

Reviewer 2 Report

In this study, the authors sought to identify novel microRNA biomarkers associated with preeclampsia through bioinformatics analysis of two publically available datasets made from small RNA sequencing of microRNAs from the placenta and plasma (GSE177049 and GSE119799 respectively). The authors identified microRNAs that were differentially expressed during preeclampsia in both the placenta and plasma, and then identified mRNA targets of 5 previously unidentified microRNAs. This useful study could help identify biomarkers of preeclampsia.

Overall, the use of 2 different datasets to identify common microRNA biomarkers of preeclampsia in different tissues is a strength of this study. The authors also did an excellent job of reviewing the literature with respect to the specific microRNAs in question. However, from reading the manuscript it is not clear whether their bioinformatics approach is sufficiently robust to identify altered microRNAs during preeclampsia. In addition, some parts of the manuscript were challenging to understand. I would recommend additional editing of the text to improve clarity for the reader.

Specific comments

(1) How were the datasets pre-processed?

The authors accessed two publically available datasets (GSE177049 and GSE119799) from Gene Expression Omnibus (GEO). GEO provides both raw sequencing files in the ‘fastq’ format, as well as read count gene expression matrices. From reading the manuscript, it is unclear whether the authors downloaded raw fastq files and performed their own pre-processing, or directly downloaded and used the read count files from GEO. There are different approaches to pre-process this type of microRNA sequencing data that can influence the resulting read counts that are used for differential expression analysis. Pre-processing of this type of data typically includes trimming of low-quality sequences, adaptor removal, alignment to a reference genome, and counting of reads that map to known microRNAs. It would be better for the authors to perform their own pre-processing on both datasets because this would ensure consistent read counts between the two datasets that were used. There are various freely available online tools to perform pre-processing, like Galaxy (https://usegalaxy.org/).

If the authors performed their own pre-processing, this should be clearly described in the methods section.

If the authors directly utilized read count files from GEO, this should be clearly explained and justified in the manuscript – including a brief description of how the two different datasets were originally pre-processed and explanation of why it is valid to make a direct comparison between them.

(2) How were the read counts normalized?

There was no description of how the read counts were normalized prior to calculating fold changes and p-values. This is important, because read count normalization drastically affects the statistical significance calculated for each microRNA. It is not valid to simply use counts per million (cpm)/transcripts per million (tpm) etc. for statistical comparisons – this type of normalization is only used for comparing expression levels between different tissues etc., and cannot be used to identify differentially expressed microRNAs. Common tools for normalizing read counts for this type of sequencing data include DESeq2 (http://bioconductor.org/packages/devel/bioc/vignettes/DESeq2/inst/doc/DESeq2.html) and EdgeR (http://bioconductor.org/packages/devel/bioc/vignettes/edgeR/inst/doc/edgeRUsersGuide.pdf), which can take into account library sizes, sequencing depth, batch effects and read distribution.

(3) Why were p-values not corrected for the false-discovery rate (FDR)?

A student’s t-test was employed to calculate p-values, with p < 0.05 considered statistically significant (lines 98-100). However, for large -omic datasets like this, it is necessary to apply a FDR correction such as the Benjamini-Hochberg correction. This will produce a more robust assessment of statistical significance. Tools like DESeq2 and EdgeR include apply a FDR correction by default when performing differential expression analysis.

(4) Principle component analysis.

Principle component analysis (PCA) is a common type of analysis used to assess sources of variation within gene expression data and could be used to visualize the effect size of preeclampsia on microRNA abundance and to identify unwanted sources of variation or batch effects present within the dataset. A PCA plot, or something equivalent, should be included because this will help the reader interpret the robustness of the results.

(5) Heatmap to visualize microRNA expression.

I would recommend the authors to include a heatmap to visualize the relative abundance of differentially expressed microRNAs across all samples within the datasets. This will help the reader to visualize expression patterns of the microRNAs and interpret the robustness of the association between altered micoRNA abundance with preeclampsia.

(6) Functional enrichment of target genes.

The authors did an excellent job of identifying mRNA targets of microRNAs by using three different databases (miRDB, RNA22, and TargetScan). While the authors did identify target genes implicated in preeclampsia through literature review, it is not clear what types of transcripts were targeted by the microRNAs and how they might be implicated in pathophysiology. I recommend the authors to perform an additional functional enrichment analysis of target genes to identify over-represented pathways and gene ontologies. There are freely available online tools to perform this type of analysis, like DAVID (https://david.ncifcrf.gov/) and EnrichR (https://maayanlab.cloud/Enrichr/).

(7) At least one of the datasets used has been previously published (GSE119799 – PMID 32871756). How does this study provide new information, compared to the work that was previously published?

(8) The specific rationale for comparing placenta and plasma microRNAs should be mentioned in the introduction.

(9) Line 122-123: “In addition, we recalculated the p-value of all of them and selected only the significant ones (p-value < 0.05)” – Why were p-values recalculated? Please clarify.

Author Response

(The authors gave the same response as above.)

Round 2

Reviewer 1 Report

I found some errors. Maybe there are more errors in this version. A careful check of this manuscript is necessary.

1.      Line 72. “Plasma” “plasma”.

2.      Figure 1 has an error. Why is the overlap number larger than those in plasma and placenta?

Author Response

Manuscript Number: genes_1862404  

“Integrated Bioinformatics Analysis Reveals Novel miRNA As Biomarkers Associated with Preeclampsia”

Dear Editor,

Thank you for your decision and for the comments about our manuscript. We have really appreciated the reviewers’comments and we believe that they have contributed a lot to improve the quality of our manuscript.

We have addressed all the reviewers’ comments and the responses are appended below.

We hope that in the revised version our manuscript is suitable for publication on Genes

Best regards

Mariarita Brancaccio on behalf of all co-authors

Response to reviewers’ comments

Reviewer 1#

I found some errors. Maybe there are more errors in this version. A careful check of this manuscript is necessary.

  1. Line 72. “Plasma” → “plasma”.
  2. Figure 1 has an error. Why is the overlap number larger than those in plasma and placenta?

Response:

Thanks for the suggestions, in agreement with you we have modified line 72 and adjusted the figure.

The number in common is greater as 298 are unique in placenta and 78 unique in plasma while 350 are those in common which are therefore found in both sets.

In addition, to improve the writing we used Grammarly software.

We hope that the manuscript is improved.

Thank you.

Reviewer 2#

The authors addressed all of my comments and I believe the manuscript has been greatly improved.

Response:

Thank you, we are happy that you enjoyed our manuscript.

Reviewer 2 Report

The authors addressed all of my comments and I believe the manuscript has been greatly improved.

Author Response

(The authors gave the same response as above.)
